# ProtGPT2 is a deep unsupervised language model for protein design

Noelia Ferruz ®[1,3] ✉, Steffen Schmidt ®[2] & Birte Höcker ®[1]

Protein design aims to build novel proteins customized for specific purposes, thereby holding the potential to tackle many environmental and biomedical problems. Recent progress in Transformer-based architectures has enabled the implementation of language models capable of generating text with human-like capabilities. Here, motivated by this success, we describe ProtGPT2, a language model trained on the protein space that generates de novo protein sequences following the principles of natural ones. The generated proteins display natural amino acid propensities, while disorder predictions indicate that 88% of ProtGPT2-generated proteins are globular, in line with natural sequences. Sensitive sequence searches in protein databases show that ProtGPT2 sequences are distantly related to natural ones, and similarity networks further demonstrate that ProtGPT2 is sampling unexplored regions of protein space. AlphaFold prediction of ProtGPT2-sequences yields well-folded non-idealized structures with embodiments and large loops and reveals topologies not captured in current structure databases. ProtGPT2 generates sequences in a matter of seconds and is freely available.

Natural language processing (NLP) has seen extraordinary advances in recent years. Large pre-trained language models have drastically transformed the NLP field and with it, many of the tools we use in our daily lives, such as chatbots, smart assistants, or translation machines. Analogies between protein sequences and human languages have long been noted by us and others[1,2]. Protein sequences can be described as a concatenation of letters from a chemically defined alphabet, the natural amino acids, and like human languages, these letters arrange to form secondary structural elements ("words"), which assemble to form domains ("sentences") that undertake a function ("meaning"). One of the most attractive similarities is that protein sequences, like natural languages, are information-complete: they store structure and function entirely in their amino acid order with extreme efficiency. With the extraordinary advances in the NLP field in understanding and generating language with near-human capabilities, we hypothesized that these methods open a new door to approach protein-related problems from sequence alone, such as protein design.

Although protein sequences and human languages are not without dissimilarities, their analogies have stimulated applying NLP methods to solve protein research problems for decades[2]. Supervised NLP methods, where the input sequences are trained jointly with their labels to produce predictive models, have been applied to various tasks, such as detecting structural similarity or predicting stability[3,4]. A remarkable collection of supervised language models applied to biomolecules is available in the BioSeq-BLM platform[5,6]. Nevertheless, since the inception of the Transformer[7], unsupervised learning, where the training occurs on unlabeled data, has emerged as a versatile tool for language modeling. Several Transformer-based models, such as TCR-BERT[8], epiBERTope[9], ESM[10], ProtTrans[11], or ProteinBERT[12], have shown to be very competitive with other methods[13,14]. Most of these models use BERT-like[15] architectures and denoising autoencoding training objectives, i.e., they are pre-trained by corrupting the input tokens in some way and trying to reconstruct the original sentence[2]. Although these models could be adjusted for generation[16], their most direct application is sequence embedding.

Another important branch of language models benefits from autoregressive training, i.e., models are trained to predict subsequent words given a context. These models, the most well-known of which

[1]Department of Biochemistry, University of Bayreuth, Bayreuth, Germany. [2]Computational Biochemistry, University of Bayreuth, 95447 Bayreuth, Germany. [3]Present address: Institute of Informatics and Applications, University of Girona, Girona, Spain. ✉e-mail: noelia.ferruz-capapey@uni-bayreuth.de

are possibly the GPT-x series[17], excel at generating long, coherent text—sometimes to the extent that much debate has been raised about their potential misuse[18]. Protein autoregressive language models, such as ProGen[19–21], RITA[22], and DARK[23] have also been studied, and show the potential of autoregressive Transformers for protein design. Motivated by these works and the ever-increasing capabilities of English-speaking models such as the GPT-x series, we wondered whether we could train a generative model to (i) effectively learn the protein language, (ii) generate fit, stable proteins, and (iii) understand how these sequences relate to natural ones, including whether they sample unseen regions of the protein space.

Here, we introduce ProtGPT2, an autoregressive Transformer model with 738 million parameters capable of generating de novo protein sequences in a high-throughput fashion. ProtGPT2 has effectively learned the protein language upon being trained on about 50 non-annotated million sequences spanning the entire protein space. ProtGPT2 generates protein sequences with amino acid and disorder propensities on par with natural ones while being "evolutionarily" distant from the current protein space. Secondary structure prediction calculates 88% of the sequences to be globular, in line with natural proteins. Representation of the protein space using similarity networks reveals that ProtGPT2 sequences explore 'dark' areas of the protein space by expanding natural superfamilies. The generated sequences show predicted stabilities and dynamic properties akin to their natural counterparts. Since ProtGPT2 has been already pre-trained, it can be used to generate sequences on standard workstations in a matter of seconds or be further finetuned on sequence sets of a user's choice to augment specific protein families. The model and datasets are available in the HuggingFace repository[24] at (https://huggingface.co/nferruz/ProtGPT2). Since protein design has an enormous potential to solve problems in fields ranging from biomedical to environmental sciences[25,26], we believe that ProtGPT2 is a timely advance towards efficient high-throughput protein engineering and design.

## Results

### Learning the protein language

The major advances in the NLP field can be partially attributed to the scale-up of unsupervised language models. Unlike supervised learning, which requires the labeling of each data point, self-supervised (or often named unsupervised) methods do not require annotated data, thus promoting the use of ever-growing datasets such as Wikipedia or the C4 Corpus[27]. Given both the growth of protein sequence databases and the lack of annotation for a significant part of the protein space, protein sequences have become great candidates for unsupervised training[4,10,11] and now offer the opportunity to encode and generate protein sequences.

To achieve this goal, we trained a Transformer[7] to produce a model that generates protein sequences. Language models are statistical models that assign probabilities to words and sentences. We are interested in a model that assigns high probability to sentences (W) that are semantically and syntactically correct or fit and functional, in the case of proteins. Because we are interested in a generative language model, we trained the model using an autoregressive strategy. In autoregressive models, the probability of a particular token or word ($w_i$) in a sequence depends solely on its context, namely the previous tokens in the sequence. The total probability of a sentence (W) is the combination of the individual probabilities for each word ($w_i$):

$$p\left(W\right) = \prod_{i}^{n} p\left(w_i|w_{<i}\right) \qquad (1)$$

We trained the Transformer by minimizing the negative log-likelihood over the entire dataset. More intuitively, the model must learn the relationships between a word $w_i$ —or amino acid—and all the previous ones in the sequence, and must do so for each sequence $k$ in dataset (D):

$$\mathscr{L}_{\text{CLM}} = -\sum_{k=1}^{D} log\, p_\theta\left(w_i^k|w_{<i}^k\right) \qquad (2)$$

To learn the protein language, we used UniRef50 (UR50) (version 2021_04), a clustering of UniProt at 50% identity. We chose this dataset versus larger versions of UniParc (such as UR100) as it was previously shown to improve generalization and performance for the ESM Transformers[10]. Uniref50's sequences populate the entire protein space, including the dark proteome, regions of the protein space whose structure is not accessible via experimental methods or homology modeling[28,29]. For evaluation, we randomly excluded 10% of the dataset sequences—these sequences are not seen by ProtGPT2 during the training process. The final training datasets contained 44.9 and 4.9 million sequences for training and evaluation, respectively. We tokenized our dataset using the BPE algorithm[30]. The final model is a decoder-only architecture of 36 layers and 738 million parameters.

Analogous to the GLUE benchmark[31]—a collection of tools that computational linguists use to evaluate language models on different tasks such as question answering or translation—we also developed a series of extrinsic tests to assess the quality of ProtGPT2-generated sequences. The following sections elaborate on how ProtGPT2 generates de novo sequences with properties that resemble modern protein space.

### Statistical sampling of natural amino acid propensities

Autoregressive language generation is based on the assumption that the probability distribution of a sequence can be decomposed into the product of conditional next-word distributions (Eq. 1). However, there is still considerable debate about the best decoding strategy to emit sequences from a model[32]. It is not uncommon that well-trained generic language models that perform well in GLUE tasks generate incoherent gibberish or repetitive text depending on the sampling procedure[32]. We briefly summarize here the most used sampling strategies for language generation that we applied in this study.

Greedy search strategy selects the word with the highest probability at each timestep. Although algorithmically simple, the generated sequences are deterministic and soon also become repetitive (Fig. 1a). Beam search tries to alleviate this problem by retaining the most probable candidates, although the resulting texts still suffer from repetitiveness and are not as surprising as those from humans, which tend to alternate low and high probability tokens[32] (Fig. 1b). Lastly, random sampling moves away from deterministic sampling by randomly picking a word out of the top-k most probable ones (Fig. 1c, d).

In a recent study, Holtzman et al.[32] investigated several sampling strategies to find the best parameters for text generation. Inspired by this work, we systematically generated sequences following different sampling strategies and parameters (Fig. 1). To assess what sampling procedure generates the most natural-like sequences, we compared the amino acid propensities of the generated set to that found in natural protein sequences (Methods). As stated by Hoffmann et al., we also observe greedy and beam search to produce repetitive, deterministic sequences, while random sampling dramatically improves the generated propensities (Fig. 1). Moreover, we also observe that high values of k are needed to generate sequences that resemble natural ones, i.e., our best results occur in the range of $k > 800$ and we specifically chose $k = 950$ in this work (Fig. 1h). As observed with other generative models[33,34], our sampling improves when applying a repetition penalty of 1.2. Consequently, we used these sampling parameters for the rest of this work.

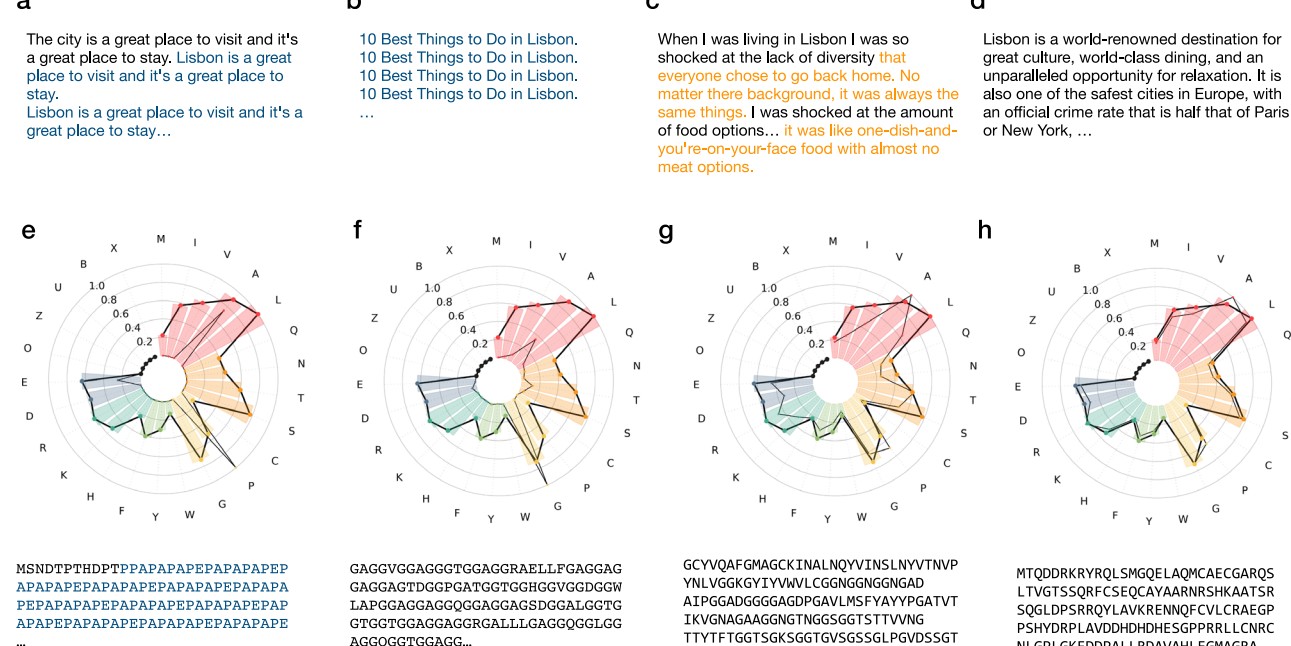

**Fig. 1 | Examples with different sampling parameters for GPT2-large after the context input: 'ten best things to do in Lisbon' (a–d) and ProtGPT2 without context (e–h).** While greedy and beam search produce repetitive sentences (**a**, **b**) and protein sequences (**e**, **f**), sampling generates creative texts, which, however, can be degenerate (**c**) or not sample natural sequence propensities (**g**) for small values of k. Larger values of k produce quality text (**d**) and sequences whose propensities match natural ones. Repetitive and degenerate text are shown in blue and orange, respectively.

## ProtGPT2 sequences encode globular proteins

In order to evaluate ProtGPT2's generated sequences in the context of sequence and structural properties, we created two datasets, one with sequences generated from ProtGPT2 using the previously described inference parameters, and the other with randomly chosen sequences from UR50. Each dataset consists of 10,000 sequences. Since ProtGPT2 was trained in an unsupervised manner, i.e., without including functional annotations, our analyses focus on validating the structural and biochemical properties of ProtGPT2 sequences.

We first studied disordered and secondary structural content in the datasets. It has been previously shown that approximately 14% of the proteins found in bacteria and archaea are disordered[28]. To this end, we ran IUPred3[35] to analyze if the ProtGPT2-generated sequences are more prone to be disordered than a set of natural sequences. Interestingly, our analysis shows a similar number of globular domains among the ProtGPT2-generated sequences (87.59%) and natural sequences (88.40%). Several methods have been reported that detect short intrinsically disorder regions[36]. Since our goal is to provide high-level comparisons of globularity and prevalent disorder across datasets, we further performed an analysis of the protein sequences at the amino acid level using IUPred3. Remarkably, our results show a similar distribution of ordered/disordered regions for the two datasets, with 79.71 and 82.59% of ordered amino acids in the ProtGPT2 and natural datasets, respectively (Table 1).

We next investigated whether the similarities in disorder are a consequence of equivalent secondary structure element content. To this end, we computed PSIPRED[37] predictions for the ProtGPT2 and natural sequence datasets. The natural sequences display alpha-helical, beta-sheet, and coil contents of 45.19, 41.87, and 12.93%, respectively. The ProtGPT2 dataset presented percentages of 48.64, 39.70, and 11.66%, respectively.

These results indicate that ProtGPT2 generates sequences that resemble globular domains whose secondary structure contents are comparable to those found in the natural space.

## Table 1 | Disorder and secondary structure predictions of the natural and ProtGPT2 dataset

|  | Natural dataset | ProtGPT2 dataset |
|---|---|---|
| **IUPred3 (globular domains)** | 88.40% | 87.59% |
| **Ordered content** | 79.71% | 82.59% |
| **Alpha-helical content** | 45.19% | 48.64% |
| **Beta-sheet content** | 41.87% | 39.70% |
| **Coil content** | 12.93% | 11.66% |

($n$ = 10,000 independent sequences/dataset).

## ProtGPT2 sequences are similar yet distant to natural ones

Proteins have diversified immensely in the course of evolution via point mutations as well as duplication and recombination. Using sequence comparisons, it is, however, possible to detect similarities between two proteins even when their sequences have significantly diverged. We wondered how related ProtGPT2 sequences are to natural ones. To this end, we utilized HHblits, a sensitive remote homology detection tool that uses profile hidden Markov models to search query sequences against a database[38]. We searched for homologs of the 10,000 sequences in ProtGPT2's dataset against the Uniclust30 database[39]. For comparison purposes, we also performed the same search with the natural dataset using the same settings. In addition, to analyze how completely random sequences would compare against ProtGPT2 ones, we also crafted a third dataset by randomly concatenating the 25 letters in the vocabulary.

Because we want to provide a quantitative comparison of the datasets' relatedness to modern protein space, we produced identity vs sequence length plots (Fig. 2). In detail, for each of the alignments found in Uniclust30, we depict the one with the highest identity and length. As a reference point in this sequence identity-length space, we use the HSSP curve[40], a boundary set to define the confidence of

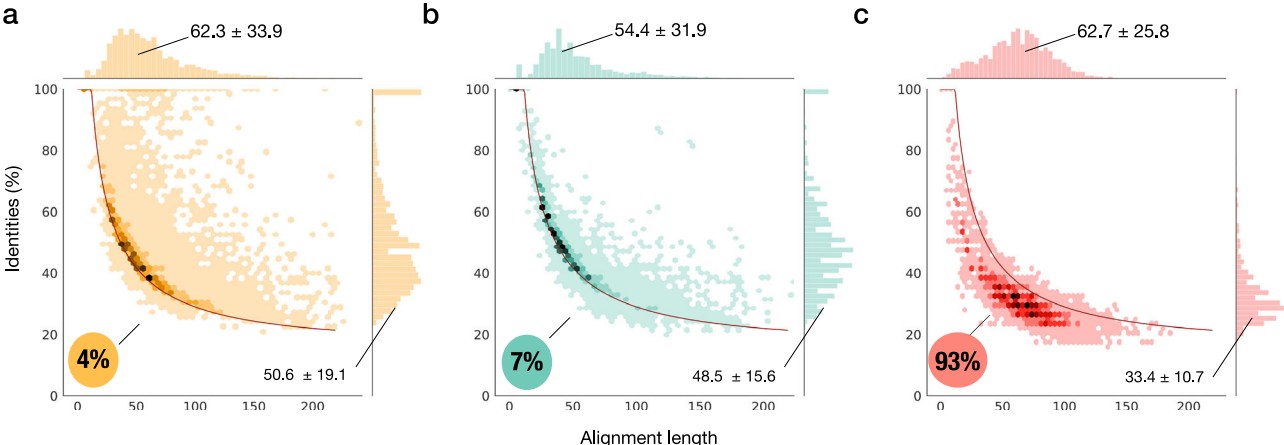

**Fig. 2 | Pairwise sequence identities vs. alignment length for each of the datasets (a: natural (yellow), b: ProtGPT2 (green), and c: random (red)) as computed with HHblits against the Uniclust30 database.** The lines depicted in red on each plot represent the HSSP curve, which we use as a reference to compare the three datasets[40]. Each plot shows a hexbin compartmentalization of the best-scoring identities and their distributions. While natural (**a**) and protGPT2 (**b**) sequences show similar percentages below the curve, 93% of the sequences in the random dataset (**c**) do not have significantly similar sequences in the Uniclust30 database. Natural and ProtGPT2 datasets show significant differences in the high-identity range (*n* = 10,000 independent sequences/dataset).

protein sequence relatedness. Proteins whose identity falls below this curve, an area known as the "twilight zone", do not necessarily have similar 3D structures nor are likely homologous. Since the sequences in the ProtGPT2 and random datasets are not the consequence of protein evolution, we use the curve as a well-known threshold to compare the datasets.

When looking at the distribution of hits above and below the curve, we observe that HHblits finds many hits in the Uniclust30 database that are related to the dataset of natural sequences (Fig. 2a). Specifically, out of the 10,000 dataset sequences, 9621 (96.2%) showed identities above the HSSP curve. Similarly, 9295 ProtGPT2-generated sequences (93%) also have counterparts in the Uniclust30 database that align above the HSSP curve (Fig. 2b). Conversely, 93% of the randomly generated sequences fall below this threshold (Fig. 2c). Despite these similar patterns for the natural and ProtGPT2 datasets, the two datasets show differences in their distribution of hits. With a one-standard-deviation range of 31.5–69.7%, the natural dataset has a higher mean identity than the ProtGPT2 set, with a range of 32.9–64.1% (Fig. 2a, b). The differences between the natural and ProtGPT2 sequence distributions are not statistically significant (*p* value <0.05 Kolmogorov–Smirnoff). However, substantial differences between the natural and ProtGPT2 datasets occur in the high-identity range (>90%). Although 365 sequences in the ProtGPT2 dataset have high-identity sequences in Uniclust30, they correspond in all cases to alignments below 15 amino acids, whereas the natural dataset displays 760 sequences over 90% with an alignment length in the one-standard-deviation range of 14.8–77.3 amino acids. These results suggest that ProtGPT2 effectively generates sequences that are distantly related to natural ones but are not a consequence of memorization and repetition.

## ProtGPT2 generates ordered structures
One of the most important features when designing de novo sequences is their ability to fold into stable ordered structures. We have evaluated the potential fitness of ProtGPT2 sequences in comparison to natural and random sequences in the context of AlphaFold predictions, Rosetta Relax scores, and molecular dynamics (MD) simulations.

AlphaFold[41,42] produces a per-residue estimate of its confidence on a scale from 0–100 (pLDDT). This score has been shown to correlate with order[43]: Low scores (pLDDT > 50) tend to appear in disordered regions, while excellent scores (pLDDT > 90) appear in ordered ones[43].

Here we produced five structure predictions per sequence. The mean pLDDT of the dataset is 63.2 when taking the best-scoring structure per sequence and 59.6 when averaging across all five predictions per sequence. Moreover, 37% of sequences show pLDDT values over 70, in agreement with other recent studies[23]. A representation of all data points is shown in Supplementary Fig. 2a. Since pLDDT scores are a proxy for structural order, we turned to the natural and random datasets to see how they compare to ProtGPT2 sequences. In agreement with previous works, 66% of the sequences in the natural dataset were predicted with pLDDT values greater than 70[43], giving an average value of 75.3 for the whole dataset (Supplementary Fig. 2b). In contrast, the predictions in the random dataset revealed a mean pLDDT value of 44, with only 7.4% of sequences with pLDDT values over 70 (Supplementary Fig. 2c).

To further validate the quality of the model, we performed Rosetta-RelaxBB runs on the three datasets[44]. Rosetta Relax performs a Monte Carlo optimization over the Rosetta energy function, which results in different backbone and rotamer conformations. Lower Rosetta Energy conformers correlate with more relaxed structures[45]. The most recent Rosetta Energy Forcefield (REF2015) strongly correlates with experimental variables such as heat capacity, density, and enthalpy[46]. This scoring function reflects the thermodynamic stability of one static protein conformation. Here we have performed Rosetta Relax experiments for the 30,000 sequences of the three datasets (Fig. 3a). A broad rule of thumb is that the total score (Rosetta Energy Units, REU) should lie between −1 and −3 per residue[47]. We observe such distribution in the natural and ProtGPT2 datasets, with averages of 1.90 and 1.73 REU/residue, respectively. As expected, the dataset of random sequences showed an average value of 0.13 REU/residue.

We further tested if ProtGPT2 sequences show similar dynamic properties as natural sequences. Proteins are dynamic entities; without their inherent flexibility, they would not be capable of interacting with other biomolecules and performing their functions in the cell[48]. To evaluate whether ProtGPT2 sequences show flexibility patterns in the same range as natural proteins, we randomly selected 12 sequences per dataset and ran three replicas of molecular dynamics (MD) of 100 ns each, totaling 108 trajectories and an aggregate time of 10.8 microseconds (Methods). To ensure that the dynamics observed during the simulations were not an artifact of different pLDDT values—and hence possible different disorder predictions—we made sure that differences among dataset-pLDDT mean values were not statistically different (Supplementary Fig. 3). The Root Mean Square Deviation means for

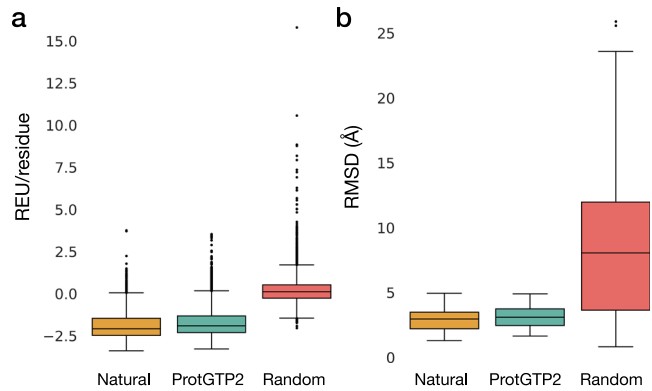

**Fig. 3 | Comparison of Rosetta and molecular dynamics calculations among the three datasets. a** Average Rosetta energy units per residue for the three datasets. AlphaFold prediction structures were used as input for the Rosetta RelaxBB protocol. 10,000 structures were run per dataset, one replica per system. **b** Root mean square deviation (RMSD) distribution for each MD dataset as computed by averaging RMSDs independently for each trajectory, represented as a boxplot. Twelve structures were simulated per dataset, three replicas per system. In both plots, the median is indicated as a black line; boxes depict the interquartile range (IQR), and whiskers represent 1.5 x IQR. Points outside this range are displayed as individual data points.

each of the trajectories in the natural and ProtGPT2 datasets resulted in average values of 2.93 and 3.12 Å, respectively (Fig. 3b). As expected, the random sequences showed significant deviations during the trajectories, with an average of 9.41 Å. While ProtGPT2 sequences showed higher values than the natural ones, the distributions are not significantly different (Mann–Whitney $U$-test, $p$ value 0.39). The results indicate that ProtGPT2 sequences might have similar dynamic properties as proteins found in nature. The complete list of the trajectories' RMSD is presented in Supplementary Figs. 4, 5.

## ProtGPT2 transcends the boundaries of the current protein space

Several studies tried to reduce the large dimensionality of protein sequences into a few discernible dimensions for their analysis. Most representation methods consist of (i) hierarchical classifications of protein structures such as the ECOD and CATH databases[49,50], (ii) Cartesian representations[51], and similarity networks[52,53]. We recently represented the structural space in a network that showed proteins as nodes, linked when they have a homologous and structurally-similar fragment in common[54] and made the results available in the Fuzzle database[55]. The network represented 25,000 domains from the seven major SCOP classes and showed that the modern known protein space has both connected and "island-like" regions.

It is implausible that evolution has explored all possible protein sequences[56]. Therefore, the challenge has been posed whether we can design proteins that populate unexplored—or dark—regions of the protein space and if, by doing so, we can design novel topologies and functions[56]. Here, we integrated the ProtGPT2 sequences into our network representation of the protein space. To this end, we generated an HMM profile for each SCOPe2.07 and ProtGPT2 sequence, compared them in an all-against-all fashion using HHsearch and represented the networks with Protlego[57]. To avoid that specific sequences with several alignments end up represented by the same node in the network, we duplicate entries with two non-overlapping alignments, as previously described[54].

The network contains 59,612 vertices and 427,378 edges, comprising 1847 components or 'island-like' clusters (Fig. 4). The major component accumulates more than half of the nodes (30,690)—a

number significantly higher than the number observed in a network produced with the same settings but excluding ProtGPT2 sequences (Supplementary Fig. 6)— strongly suggesting that ProtGPT2 generates sequences that bridge separate islands in protein space. We select six examples across different areas of the network from topologically different SCOPe classes to showcase ProtGPT2 sequences at the structural level (Fig. 4). In particular, we report an all-β (**751**), two α/β (**4266**, **1068**), one membrane protein (**4307**), an α + β (**486**) and all-α (**785**) structures. These structures illustrate ProtGPT2's versatility at generating de novo structures. For each case, we searched the most similar protein structure found in the PDB database using FoldSeek[58]. ProtGPT2 generates well-folded all-β structures (**751**, **4307**), which despite recent impressive advances[59], have for long remained very challenging[60]. ProtGPT2 also produces membrane proteins (**4307**), which pose a difficult target for protein design due to the challenges at specifying structure within the membrane and the laborious experimental characterizations[61]. Besides the generation of natural fold representatives, ProtGPT2 also produces previously unreported topologies. For example, we report protein **4266**, whose topology does not match any of the currently reported structures in the PDB, with a low DALI Z-score of 5.4 and an RMSD of 3.0 Å to PDB 5B48 over 67 residues (identity 9%).

Nevertheless, possibly the most remarkable property of ProtGPT2 sequences is their significant deviation from all previously designed de novo structures, which often feature idealized topologies with loops and minimal structural elements. De novo proteins have the advantage of not carrying any evolutionary history and are thus amenable as a scaffold for virtually any function, but in practice, the lack of embodiments and longer loops hamper the design of crevices, surfaces, and cavities—necessary for the interaction with other molecules and function realization. ProtGPT2 sequences resemble the complexity of natural proteins, with multifaceted surfaces capable of allocating interacting molecules and substrates, thus paving the way for functionalization. In Fig. 4, we show structures **486** and **1060**, two examples of such complex structures. In particular, **1068** shows a TIM-barrel fold, a topology which to date has met impressive success in de novo design[62–64], but whose idealized structure has nevertheless proven challenging to extend via additional secondary elements and longer loops[65,66].

## Preserved functional hotspots

Visual inspection of the structural superimposition of the best hits found with FoldSeek revealed several instances where the sidechains of ligand-interacting residues are conserved. Two examples are shown in Fig. 5. The natural structure most similar to sequence **357** (Fig. 5a) corresponds to PDB code 1X0P (chain A), a blue-light sensor domain that binds FAD. When superimposing the structures, we observe that **357** has retained the sidechain binding hotspots, with three residues identical (D169, Q150, and N131) and two different but capable of forming the same interactions, Lysine at position R165 and Histidine at position K127. Sequence **475** (Fig. 5b) is most similar to PDB code 5M1T (chain A), a phosphodiesterase that folds into a TIM-barrel and binds to the bacterial second messenger cyclic di-3′,5′-guanosine monophosphate (PDB three-letter code C2E). Out of the five sidechain-interacting residues, the ProtGPT2 sequence preserves three residues (Q455, R473, and E469), and includes one substitution for another residue capable of hydrogen-bonding (aspartic acid for Q513). It is remarkable to note that ProtGPT2 has generated these sequences in a zero-shot fashion, i.e., without further finetuning in these two particular folds. These results have impactful consequences for protein engineering because ProtGPT2 appears to preserve binding positions in the generated sequences, despite the low identities (31.1 and 29.2% for 357 and 45, respectively), and can be used to augment the repertoires of specific folds and families.

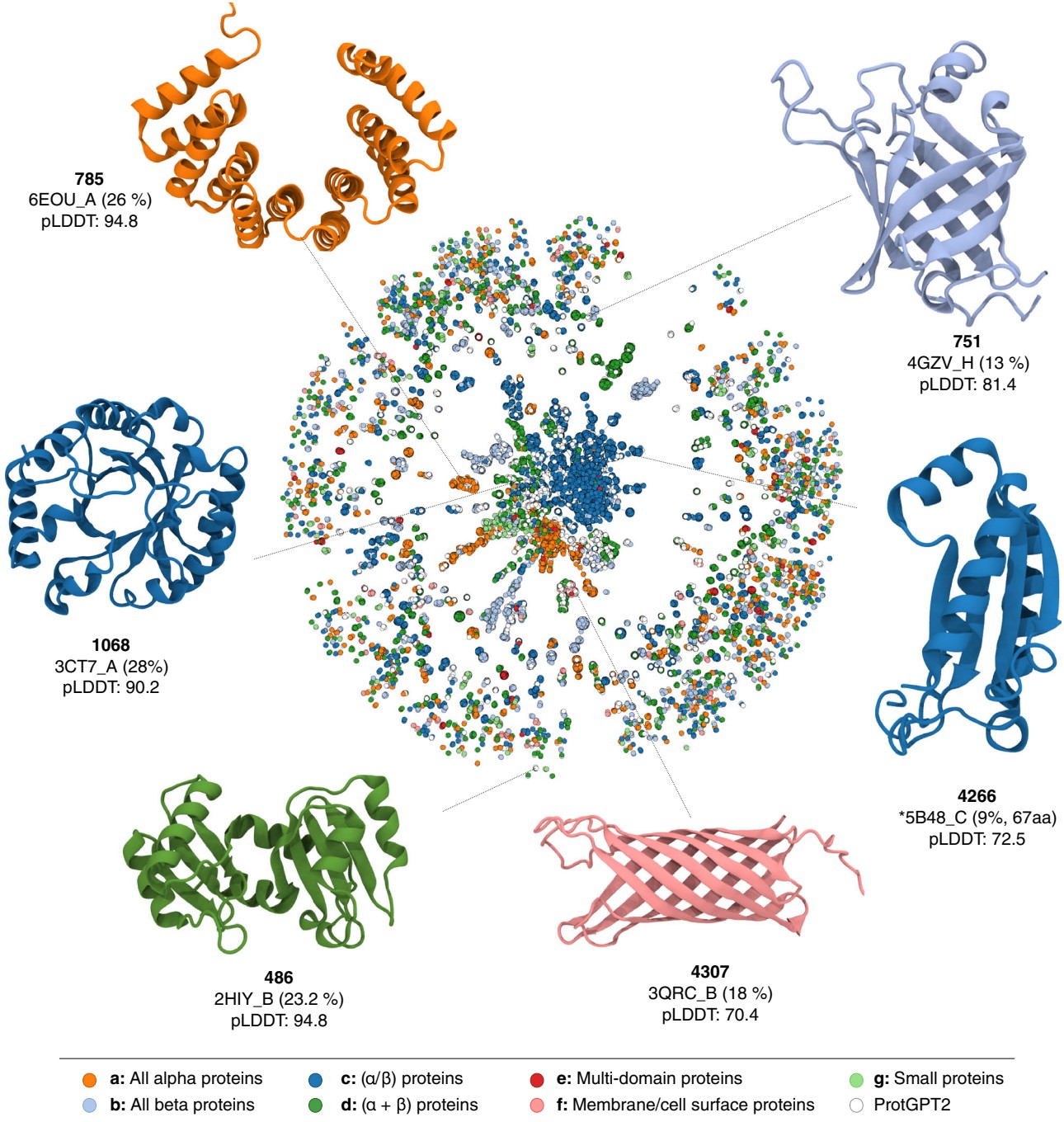

**Fig. 4 | An overview of the protein space and examples of proteins generated by ProtGPT2.** Each node represents a sequence. Two nodes are linked when they have an alignment of at least 20 amino acids and 70% HHsearch probability. Colors depict the different SCOPe classes, and ProtGPT2 sequences are shown in white. As examples, we select proteins of each of the major five SCOP classes: all-β structures (751), α/β (4266 and 1068), membrane protein (4307), α+β (486), and all-α (785). The selected structures are colored according to the class of their most similar hit. The structures were predicted with AlphaFold, and we indicate the code of the most similar structure in the PDB as found by FoldSeek[58], except for protein 4266, where no structures were found.

## Discussion

The design of de novo proteins harnessing artificial intelligence methods has been meeting incredible success in the last 2 years[10,67,68]. Motivated by the unprecedented advances in NLP, we have implemented a generative language model, ProtGPT2, which has effectively learned the protein language. ProtGPT2 can generate sequences that are distantly related to natural ones and whose structures resemble the known structural space, with non-idealized complex structures. Since ProtGPT2 has been trained on the entire sequence space, the sequences produced by the model can sample any region, including

the dark proteome and areas traditionally regarded as very challenging in the protein design field, such as all-β structures and membrane proteins. Visual superimposition of ProtGPT2 proteins with distantly related natural protein structures reveals that ProtGPT2 has also captured functional determinants, preserving ligand-binding interactions. As the design of artificial proteins can solve many biomedical and environmental problems, we see extraordinary potential in our protein language model. ProtGPT2 designs fit globular proteins in a matter of seconds without requiring further training on a standard workstation. ProtGPT2 can be conditioned towards a particular family, function, or

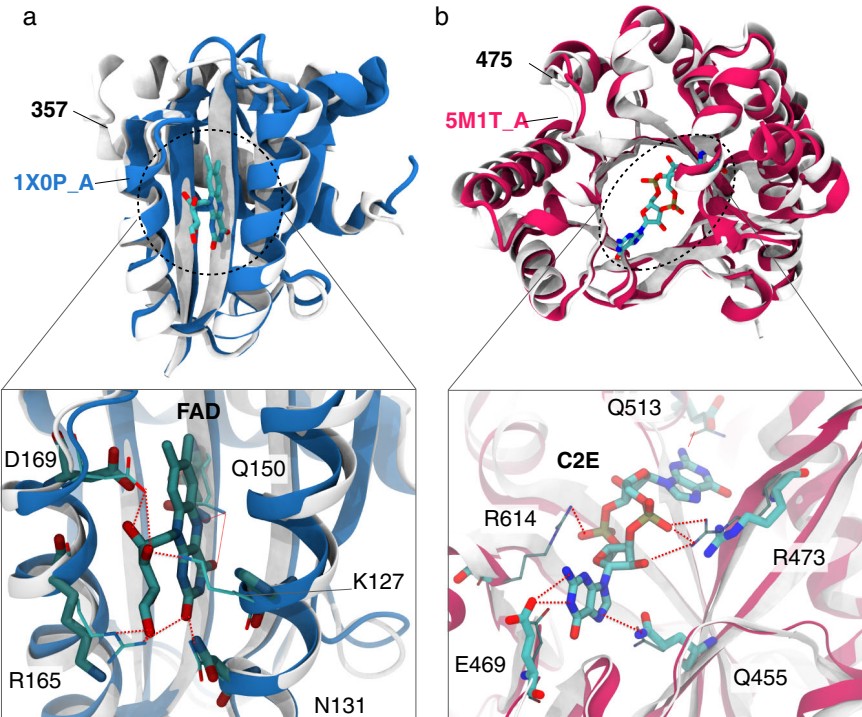

**Fig. 5 | Superimposition of the predicted structures for sequences 357 and 475 and the respective top scoring proteins in FoldSeek. a** Structural alignment of 357 with pdb 1X0P (chain A, blue). Shown are five residues in 1X0P that interact via their sidechains with the ligand FAD. Of these, three are identical in **357**, and another two correspond to substitutions to the same amino acid type (R165 to

lysine and Q150 to histidine). **b** Structural alignment of **475** with pdb 5M1T (chain A) depicting five sidechain-interacting residues with ligand C2E. All amino acids in **475** are conserved except for residue R614, which was substituted by a glycine. The PDB structures are shown in color with their sidechains in a thinner representation.

fold by finetuning the model on a set of sequences of a user's choice. In this context, ProtGPT2 will enable the screening for proteins with similarities to natural proteins in order to improve, fine-tune or alter a specific biochemical function of a natural protein. Large-scale screening of ProtGPT2-designed protein libraries might identify proteins with folds not captured in structural databases and functions that have no related counterpart in the natural space. ProtGPT2 constitutes a big step forward towards efficient protein design and generation, and lays the groundwork for future experimental studies exploring the structural and functional parameters of designed proteins, and their subsequent real-world applications. Future efforts include the inclusion of conditional tags, which will enable the controlled generation of specific functions.

## Methods

### Vocabulary encoding
We use a BPE[30] tokenizer to train the vocabulary of our dataset. BPE is a sub-word tokenization algorithm that finds the most frequently used word roots, ensuring better performance than one-hot tokenization and avoiding the out-of-vocabulary problem. Given the size of Uniref50, we used Swiss-Prot (2021_04) containing >0.5 M sequences to train our tokenizer. Following the training strategy of GPT2[17], our final vocabulary contained 50,256 tokens that correspond to the most widely reused oligomers in protein space, with an average size of four amino acids per token (Supplementary Fig. 1). Learned positional embeddings were used as in the original GPT2.

### Dataset preparation
We took Uniref50 version 2021_04 as the dataset for training, containing 49,874,565 sequences. 10% of the sequences were randomly selected to produce the validation dataset. The final training and validation datasets contained 44.88 and 4.99 million sequences,

respectively. We produced two datasets, one using a block size of 512 tokens, and another one with 1024 tokens. The results shown in this work correspond to a model trained with a block size of 512 tokens.

### Model pre-training
We use a Transformer decoder model as architecture for our training which processes input sequences tokenized with a BPE strategy. The model uses during training the original dot-scale self-attention as introduced by ref. 7. The model consist of 36 layers with a model dimensionality of 1280. The architecture matches that of the previously released GPT2-large Transformer[17], which was downloaded from HuggingFace[24]. Model weights were reinitialized prior to training. The model was optimized using Adam ($\beta_1 = 0.9$, $\beta_2 = 0.999$) with a learning rate of 1e-03. For our main model, we trained 65,536 tokens per batch (128 GPUs × 512 tokens). A batch size of 8 per device was used, totaling 1024. The model trained on 128 NVIDIA A100s in 4 days. Parallelism of the model was handled with DeepSpeed[69].

### Model inference
We systematically sampled sequences using our main model using different inference parameters. In particular, we varied the repetition penalty from a range of 1.1 to 3.0 at each 0.1 units, top_k from 250 to 1000 sampling every 50 units, and a top_p from 0.7 to 1.0 with a window of 0.05 units. 100 sequences were produced for each sampling parameter set and the frequency of their amino acids compared to natural sequences. We observed which parameters produced fewer differences in the set of the seven most common amino acids in natural sequences. We also explored the beam search algorithm for beams in the range 50 to 100 using a window of 1 unit but it produced worse matches in all cases. To determine amino acid frequencies in natural sequences for comparison to ProtGPT2 samples, we

randomly picked 1 million sequences from the Uniref50 dataset. The best matching parameters were further downsampled with finer windows and their frequencies compared with radar plots, as shown in Fig. 1 in the main text. The best performing parameters in our dataset were top_k 950, repetition penalty of 1.2, and default temperature and top_p values of 1.

## Sequence dataset generation

Three sequence datasets were produced to compare their properties. The ProtGPT2 dataset was generated by sampling 1000 batches of 100 sequences, each with the selected inference parameters and a window context of 250 tokens. This step produced 100,000 sequences. We filtered from this set those sequences whose length had been cut due to the window context, giving a total of 29,876 sequences. From this set, we randomly selected 10,000 sequences. Their average length is 149.2 ± 50.9 amino acids. The natural dataset was created by randomly sampling 100,000 sequences from Uniref50. 10,000 of these sequences were further chosen to ensure their average and standard deviation lengths matched that of the ProtGPT2 dataset sequences. The random dataset was created by concatenating the 25 amino acids that appear in UniRef50, which includes the 20 standard amino acids and other IUPAC codes such as "X", "B", "U", "O", and "Z", by randomly concatenating them into sequences with a length taken from a normal distribution between 5 and 267 amino acids.

## Homology detection

Each sequence in the three 10k datasets was searched for similarity against the PDB70 and uniclust30 databases using HHblits[70]. We used the Uniclust30 database version 2018_08 and the pdb70 version 2021_04. As HHblits produces a list of alignments we selected all those over the HSSP curve as possible matches, and from these, selected the largest alignment. Thus, for each sequence in each dataset, the longest and the highest identity scoring alignment was selected and represented in Fig. 2.

## Disorder prediction

IUPred3 was run on ProtGPT2 and natural datasets using all three possible options to detect shorter ("short") or longer ("longer") unstructured regions, as well as structured regions ("glob")[35]. Ordered content was determined with the "short" option. The output of the "glob" analysis also reports if any structured, globular domain was found, as shown in Table 1. We ran secondary structure prediction using PSIPRED v4.0 for each sequence in natural and ProtGPT2 datasets[37]. The alignments of the abovementioned HHblits searches were used as multiple sequence alignments. We computed the percentages for each secondary element by dividing the number of amino acids with a certain prediction by the total number of amino acids with a confidence value of 5 or more.

## AlphaFold2 structure prediction

We predicted five structures for each sequence in the ProtGPT2 dataset using AlphaFold ColabFold batch v1.2[41].

## Network construction

Sequences in the ProtGPT2 and SCOP 2.07 filtered at 95% datasets were joined. For each sequence, we produced a multiple sequence alignment (MSA) using HHblits against the database Uniclust 2018_08. Hidden Markov model profiles were produced for each MSA using HHblits[70], and an all-against-all search for each profile was performed using HHsearch[38]. The network was constructed by representing every sequence as a node, and linking two nodes whenever they have an alignment of at least 20 amino acids with 70% HHsearch probability. Extensive details on the all-against-all comparison and network construction, and tools to generate the networks can be found in our

previous works Fuzzle[54,55] and Protlego[57]. Detection of similar topologies was determined with FoldSeek[58].

## Molecular dynamics simulations

Simulation systems were built and run with the software HTMD[71]. In all cases, systems comprised solvated all-atom cubic boxes. Simulation boxes consisted of a protein centered at the origin of coordinates and explicit solvent molecules and neutralizing NaCl ions were added to each box. The Amber 19SB forcefield was used[72]. Three replicas were constructed per sequence. All systems were minimized, equilibrated, and run with ACEMD[73] using default parameters: each system was minimized and relaxed under NPT conditions for 1 ns at 1 atm and 300 K using a time-step of 4 fs, rigid bonds, cutoff of 9 Å, and PME for long-range electrostatics. Heavy protein and ligand atoms were constrained by a 10 kcal/mol/Å2 spring constant. Production simulations were run in the NVT ensemble using a Langevin thermostat with a damping of 0.1 ps⁻¹ and a hydrogen mass repartitioning scheme to achieve timesteps of 4 fs[74].

## Rosetta calculations

Rosetta Relax runs were produced with the Rosetta Software Suite v3.12[44] using as input structure the best-scoring prediction from AlphaFold.

## Reporting summary

Further information on research design is available in the Nature Research Reporting Summary linked to this article.

## Data availability

The model weights are publicly available in the HuggingFace repository: https://huggingface.co/nferruz/ProtGPT2 and Zenodo: https://doi.org/10.5281/zenodo.6796843 [https://zenodo.org/record/6796843#.YswB9XbMIVA]. The dataset for training is available at: https://huggingface.co/datasets/nferruz/UR50_2021_04. The three sequence datasets in this work are available at: https://huggingface.co/datasets/nferruz/dataset_fastas. The AlphaFold predictions for the three datasets are available at https://huggingface.co/datasets/nferruz/dataset_alphafold. The Uniref50 original database version 21_04 is available at https://ftp.uniprot.org/pub/databases/uniprot/previous_releases/release-2021_04/. The Uniclust30 database version 2018_08 is available at http://gwdu111.gwdg.de/~compbiol/uniclust/2018_08/uniclust30_2018_08_hhsuite.tar.gz.

## Code availability

The model was trained with the HugginFace transformers Trainer version 4.14.1. The code and documentation are available here: https://huggingface.co/docs/transformers/main_classes/trainer.

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

## Acknowledgements
The authors gratefully acknowledge the scientific support and HPC resources provided by the Erlangen National High-Performance Computing Center (NHR@FAU) of the Friedrich-Alexander-Universität Erlangen-Nürnberg (FAU) under an early-access NHR project. NHR funding is provided by federal and Bavarian state authorities. NHR@FAU hardware is partially funded by the German Research Foundation (DFG) —440719683. We thank Thomas Zeiser for his considerate support and Surbhi Dhingra for feedback on the manuscript. N.F. acknowledges support from an AGAUR Beatriu de Pinós MSCA-COFUND Fellowship (project 2020-BP-00130). The authors thank funding from the German Research Foundation (DFG) - 491183248 and the Open Access Publishing Fund of the University of Bayreuth.

## Author contributions
N.F conceived the work, trained the model, analyzed the data, and wrote the manuscript. S.S produced the IUPred3 disorder predictions and analysis and wrote the manuscript. B.H analyzed the data and wrote the manuscript. The three authors discussed the results and supervised the work.

## Funding

## Competing interests
The authors declare no competing interests.
