## [Peer Review File · Nature Communications]

Reviewers' Comments:

Reviewer #1:

Remarks to the Author:

Key results:

In the paper, the authors present an autoregressive, deep learning language generation model for protein sequences, based on the GPT2 architecture and trained on UniRef50 (~50 M sequences, 50% identity), i.e., a "ProtGPT". They use BPE tokenization instead of character-level wise for tokenizing and generating the sequences. The authors evaluate generated "novel" sequences based on AA similarity and predicted disorder, and predicted 2D structure (According to AlphaFold2) having similar global frequencies to natural proteins. They made their model available using the popular HF library. The presentation (visually and conceptually) is attractive. The logic and the protocol are well presented.

Data and methodology

The data and modelling approaches used are all valid and justified. The k-mer tokenization makes much more sense than character level for this sort of problem, and using UniRef50 is a good way to save on compute costs while getting generalizability, even if it would lose on learning of many "minute" variants belonging to the same family.

Other related published papers using Natural Language generation deep-learning architectures for biological/protein sequence generation papers, both for the domain of antigens and the immune system including The TCR-BERT (doi:10.1101/2021.11.18.469186), the EpiBERTope (doi: 10.1101/2022.02.27.481241), and in general, across all proteins: ProGen from Socher lab (doi: 10.1101/2020.03.07.982272); ProtTrans / ProtXNet from Rost lab (doi: 10.1101/2020.07.12.199554) and the design 3D-based trRosetta approach from baker lab (doi: 10.1038/s41586-021-04184-w). Model architectures that are specialized to the domain of proteins are available in some projects such as ProteinBERT, epigenomicBERT, Berger lab (doi: 10.48550/arXiv.1902.08661).

Validity:

The validation of the candidate generated sequences is also not enough in my view to confidentially establish this as a tool, without further validation. In Socher paper (ProGen), it performed an in-depth analysis of stability of candidates in silico, while others (Cell) synthesized candidates and checked the function. Ideal would be synthesis of some candidate protein and experimental validation, even if it is a "simple" case, example being, a variant of a known protein, checking antibody binding, a peptide, expression and such. I do not consider experimental validation a sole criterion, but I do feel that the current level of evidence is insufficient to show a clear improvement, or that generated candidates (with low sequence similarity to existing sequences) necessarily maintain function.

An alternative would be to show if the model has better performance as a pretrained feature extractor/transfer learning model, for existing in-silico protein benchmarks. A known benchmark is provided by Song lab (Tasks Assessing Protein Embeddings (TAPE) which is a set of tasks across different domains of protein biology. The TAPE was used successfully in the case of ProteinBERT (Bioinformatics, doi: 10.1093/bioinformatics/btac020) and ProtTrans. There may be other 'validation' options that are sufficient so long as they show some validated evidence for the model generating functional, novel proteins or better learning protein statistical properties for modelling purposes.

Significance:

The availability of the model with trained weights is very useful and a possible boon to the community. At the stage of research in NLP-proteins where there are many more DL efforts for protein design and NLP in proteins, more compelling proof of the model working well are needed. However, it being freely available is a major advantage (assuming this is sufficient for the journal), especially given the substantial compute costs of such language models (especially as ProGen's model is not available).

Reviewer #2:

Remarks to the Author:

The authors proposed an auto-regression based protein generative language model (ProtGPT2) for the de novo design of proteins. Although the topic deserves attention, the motivation, assessment and presentation of this manuscript should be improved.

Major problems

1. This motivation of this study should be clearly discussed. Some pre-trained protein language models have been proposed, such as Bepler T. and Berger B. (2019), Bepler T. and Berger B. (2021), ProtTrans and ProGen, and the ProtGPT2 is similar with protGen, the generative-based language model. Moreover, why are the auto-encoding base language models, such as BERT and other auto-encoding language models not suitable for the task of de novo protein design? The differences between the existing models and the current language model needs to be illustrated.
2. Recently, some platforms have been proposed to generate different biological sequence language models, such as BioSeq-BLM, BioSeq-Analysis2.0, etc. Did the authors try these models for their current task? Or at the very least, the authors should discuss these recent language models for bioinformatics.
3. Method lacks the comparisons with some important works. The performance comparison between the existing auto-regression based protein generative language model (Bepler T. and Berger B. (2019) and ProGen) should be explored (see comment 1).
4. What manner did the authors employ to initialize the protein vocabulary (BPE) encoding? Did the authors use position encoding in the Transformer model? If not, please explain why?
5. Table 1 reported the disorder and secondary structure predictive results of generated sequences and natural sequences. However, in order to avoid the prediction bias of the predictor, the predictive results on the generated sequences and natural sequences should be reported to perform the comparison, in other words, the natural content values in Table 1 should be replaced by the predictive results on natural sequences.

Minor

1. Figure 1: the examples of text generation in Fig1 (A-C) do not match with that in Fig1 (E-G). The Amino acid distributions of one sequence in Fig1 (E-H) are unconvincing, but the calculations on the entire development set are more acceptable.
2. Section "ProtGPT2 sequences are similar yet distant to natural ones": "randomly concatenating the 25 letters in the vocabulary." What are the 25 letters and how to control the length of randomly concatenating sequences?

We appreciate the comments and suggestions from all reviewers. We believe these suggestions have improved the quality of our manuscript immensely and hope that our revision has addressed their concerns. We show our answers in blue.

REVIEWER COMMENTS

Reviewer #1 (Remarks to the Author):

Key results:

In the paper, the authors present an autoregressive, deep learning language generation model for protein sequences, based on the GPT2 architecture and trained on UniRef50 (~50 M sequences, 50% identity), i.e., a "ProtGPT". They use BPE tokenization instead of character-level wise for tokenizing and generating the sequences. The authors evaluate generated "novel" sequences based on AA similarity and predicted disorder, and predicted 2D structure (According to AlphaFold2) having similar global frequencies to natural proteins. They made their model available using the popular HF library. The presentation (visually and conceptually) is attractive. The logic and the protocol are well presented.

We are thankful the reviewer finds our manuscript well presented.

Data and methodology:

The data and modelling approaches used are all valid and justified. The k-mer tokenization makes much more sense than character level for this sort of problem, and using UniRef50 is a good way to save on compute costs while getting generalizability, even if it would lose on learning of many "minute" variants belonging to the same family. Other related published papers using Natural Language generation deep-learning architectures for biological/protein sequence generation papers, both for the domain of antigens and the immune system including The TCR-BERT (doi:10.1101/2021.11.18.469186), the EpiBERTope (doi: 10.1101/2022.02.27.481241), and in general, across all proteins: ProGen from Socher lab (doi: 10.1101/2020.03.07.982272); ProtTrans / ProtXNet from Rost lab (doi: 10.1101/2020.07.12.199554) and the design 3D-based trRosetta approach from baker lab (doi: 10.1038/s41586-021-04184-w). Model architectures that are specialized to the domain of proteins are available in some projects such as ProteinBERT, epigenomicBERT, Berger lab (doi: 10.48550/arXiv.1902.08661).

Thank you for the extensive literature. We realized some of these references were missing, we have now incorporated them in the Introduction section where appropriate.

Validity:

The validation of the candidate generated sequences is also not enough in my view to confidentially establish this as a tool, without further validation. In Socher paper (ProGen), it performed an in-depth analysis of stability of candidates in silico, while others (Cell) synthesized candidates and checked the function. Ideal would be synthesis of some candidate protein and experimental validation, even if it is a "simple" case, example being, a variant of a known protein, checking antibody binding, a peptide, expression and such. I do not consider experimental validation a sole criterion, but I do feel that the current level of evidence is insufficient to show a clear improvement, or that generated candidates (with low sequence similarity to existing sequences) necessarily maintain function. An alternative would be to show if the model has better performance as a pretrained feature extractor/transfer learning model, for existing in-silico protein benchmarks. A known benchmark is provided by Song lab (Tasks Assessing Protein Embeddings (TAPE) which is a set of tasks across different domains of protein biology. The TAPE was used successfully in the case of ProteinBERT (Bioinformatics, doi: 10.1093/bioinformatics/btac020) and ProtTrans. There may be other 'validation' options that are sufficient so long as they show some validated

evidence for the model generating functional, novel proteins or better learning protein statistical properties for modelling purposes.

We agree that validating the quality of the sequences is important. Since ProtGPT2 is an autoregressive model, we had not deemed comparisons with denoising autoencoders – TAPE benchmark, ESM, ProteinBERT, etc.- appropriate due to the different training objectives. Besides, since our primary objective is protein design, we believe that characterization of the generated sequences' properties across different dimensions is a more appropriate benchmark for our model.

We have now included a section validating the sequences at three levels:

1. We have run extensive molecular dynamics simulations, totalling over 10,000 nanoseconds of aggregate simulation time.
2. We have computed Rosetta Relax scores for the 30,000 sequences of the three datasets in the manuscript.
3. We have extended our AlphaFold prediction to the natural and random sequence datasets (20,000 structures).

Taken together, our results show that the sequences generated by ProtGPT2 show dynamic properties, stability scores, and disorder tendencies in line with natural sequences despite being novel and distant from the protein space.

Since ProtGPT2 was not trained to model functional information, we have not included functional predictions at this stage. We plan to train another model with functional tags in the future.

Significance:

The availability of the model with trained weights is very useful and a possible boon to the community. At the stage of research in NLP-proteins where there are many more DL efforts for protein design and NLP in proteins, more compelling proof of the model working well are needed.

However, it being freely available is a major advantage (assuming this is sufficient for the journal), especially given the substantial compute costs of such language models (especially as ProGen's model is not available).

We are delighted that the model is considered a possible boon to the community. We also believe that releasing such large models is ultimately beneficial for the entire field.

Reviewer #2 (Remarks to the Author):

The authors proposed an auto-regression based protein generative language model (ProtGPT2) for the de novo design of proteins. Although the topic deserves attention, the motivation, assessment and presentation of this manuscript should be improved.

Major

1. This motivation of this study should be clearly discussed. Some pre-trained protein language models have been proposed, such as Bepler T. and Berger B. (2019), Bepler T. and Berger B. (2021), ProtTrans and ProGen, and the ProtGPT2 is similar with protGen, the generative-based language model. Moreover, why are the auto-encoding base language models, such as BERT and other auto-encoding language models not suitable for the task of de novo protein design? The differences between the existing models and the current language model needs to be illustrated.

Thanks for pointing this out. We have updated the Introduction to discuss the differences and similarities to other models more straightforwardly. More specifically, the Bepler T and Berger B (2019) manuscript is an exciting application of language models to predict global structural similarity and contact prediction, ultimately outperforming established tools such as TMalign for classifying domains in the SCOPe database. This work, however, uses a different architecture (biLSTMs) and does not explore protein design, which is our primary goal. Bepler T. and Berger B. (2021) reviews recent language models focusing on protein prediction problems and introduces another biLSTM that incorporates structural knowledge. Their architecture differs from the one we used (Transformer), and their immediate goal is not protein design.

ProtTrans is a collection of the most relevant Transformer architectures released in the last few years applied to the protein Corpora, whose majority of models are trained with a masked autoencoding objective and thus cannot directly perform text (or sequence) generation. Two of the models were trained autoregressively (Transformer-XL, and XLnet) but offered slightly different training objectives and architectures and more importantly, did not explore the possibility of generating proteins. ProGen is the most similar work to ours, but it was trained with control tags, and the model has not been released. Our approach is also different to the latter in that we are interested in understanding how our model 'speaks' the protein language. Namely, we are focused on 1) understanding how the model interprets the protein language by comparing the properties of generated and natural sequences and 2) exploring -if possible- unseen yet plausible regions of the protein space.

To clarify these differences we have considerably updated the Introduction section. Some of these references were mentioned in our work, but we have completed those missing. We hope the basis to perform this work are now well-motivated to the readers.

2. Recently, some platforms have been proposed to generate different biological sequence language models, such as BioSeq-BLM, BioSeq-Analysis2.0, etc. Did the authors try these models for their current task? Or at the very least, the authors should discuss these recent language models for bioinformatics.

BioSeq-BLM is a webserver that collects an impressive collection of different NLP tools applied to DNA, RNA, and protein sequences. We tried extensively to design sequences with the server -one of the models is a Transformer- but the server suggests to use the standalone version for this purpose. However, the standalone version provides solely the framework to train user-specific models from scratch. Such notion is similar to the HuggingFace framework we have used to train our model, although these tools make use of a larger set of NLP architectures. The mentioned tools cannot be used to generate sequences unless a model is trained such as we have done in this publication. The references are included in the text.

3. Method lacks the comparisons with some important works. The performance comparison between the existing auto-regression based protein generative language model (Bepler T. and Berger B. (2019) and ProGen) should be explored (see comment 1).

We cannot compare against the ProGen model because the model is not made publicly available. Regarding the Bepler T. & Berger B. (2019) model is available at <https://github.com/tbepler/protein-sequence-embedding-iclr2019>, more recently a newer version has been made available at <https://github.com/tbepler/prose>. These models are biLSTMs, i.e., a different architecture to the one we have trained, and produce an embedding for each position of the sequence. This concept is similar to the ESM Transformers, which were trained with a masking autoencoding objective and produce a sequence embedding that can be coupled to downstream tasks. Since the model does not generate sequences, we cannot perform a comparison against ProtGPT2.

We, however, fully understand the reviewer's concerns on the need to validate our model further, and we agree that our first manuscript lacked validation of the generated sequences in terms of stability and fitness. For this reason, we have extended our manuscript with validations of the generated sequences in different contexts. We now provide tests at three levels. First, we performed high-throughput molecular dynamics, totalling over 10,000 nanoseconds of aggregate simulation time. Second, we computed Rosetta Relax scores for the 30,000 sequences in the manuscript and compared the three datasets. Lastly, we extended our AlphaFold2 prediction to the 20,000 sequences from the natural and random sequence datasets.

4. What manner did the authors employ to initialize the protein vocabulary (BPE) encoding? Did the authors use position encoding in the Transformer model? If not, please explain why?

We used BPE encoding in the same manner as GPT2 to keep the two training specifications consistent. Likewise, we also use learned position embeddings such as the original GPT2 work. We have updated this information in the Methods section of the manuscript.

5. Table 1 reported the disorder and secondary structure predictive results of generated sequences and natural sequences. However, in order to avoid the prediction bias of the predictor, the predictive results on the generated sequences and natural sequences should be reported to perform the comparison, in other words, the natural content values in Table 1 should be replaced by the predictive results on natural sequences.

Yes, this is a good point and is in fact what we are doing. We computed these variables for the two datasets, the values from the natural dataset do not come from literature searches. This way, as you suggest, we avoid the prediction bias of the predictor.

Minor

1. Figure 1: the examples of text generation in Fig1 (A-C) do not match with that in Fig1 (E-G). The Amino acid distributions of one sequence in Fig1 (E-H) are unconvincing, but the calculations on the entire development set are more acceptable.

Figure 1 shows known problems of the different generation approaches. Only sampling techniques with different top k and p parameters yield coherent text in English (Fig D). Greedy and Beam search algorithms tend to produce repetitive text (Fig. 1A, B). We observe that this is the case for proteins, too, as shown in the amino acid distributions (Fig. E-H). Only the sequence shown in Fig. H is supposed to follow natural properties.

2. Section “ProtGPT2 sequences are similar yet distant to natural ones”: “randomly concatenating the 25 letters in the vocabulary.” What are the 25 letters and how to control the length of randomly concatenating sequences?

The training set (Uniref50) contains sometimes IUPAC codes for special amino acids (e.g. ‘U’ for selenocysteine) or ambiguous amino acids (e.g. ‘B’ for aspartic acid or asparagine) in a sequence. Hence, the data set contains additional letters than the typical 20 letters for the amino acids. Regarding the length, we generated 10,000 sequences of any length randomly picked from a normal distribution between 5 and 267 amino acids. This way, we ensured that we had equivalent proportions of sequence lengths across the three datasets. This process is explained in the Methods section. We wrote: *‘The random dataset was created by concatenating the 25 amino acids that appear in UniRef50, which includes the 20 standard amino acids and other IUPAC codes such as ‘X’, ‘B’, ‘U’, ‘O’, and ‘Z’, by randomly concatenating them into sequences with a length taken from a normal distribution between 5 and 267 amino acids.’*

Reviewers' Comments:

Reviewer #1:

Remarks to the Author:

Review (Rebuttal)

In 'design' papers, a legitimate norm is to use some of the prediction for experimental validation (as in many excellent studies, including Anishchenko et al. (2021, on DL hallucination) and others. With a lack of experimental validation (which is 'hard 'but convincing), generalization of the presented tools is done by setting some statistical evidence for "success". Such measures seek the shortest distance between unseen, new, and known sequences and structures. Such validation is, of course, important and a critical step towards improving models in the protein design field.

The main shortcomings of the original manuscript were the lack of validation and the somewhat deficient novelty. In this version, the authors expanded their tests by testing "novel" sequences statistics versus predicted features (e.g., 2D structure, as presented by AlphaFold2) . In the current version, the authors addressed the major comments by providing new set of validations (not for protein functions but for a mode of structurally oriented tasks). The addition of Rosetta Relax scores (RRS) and the extension of AlphaFold prediction to natural and random sequences (20k) are critical additions to improve the manuscript. The other addition concerns testing the protein dynamics (using MD simulations). Also, it suffered from some missing references to background knowledge (that is changing very fast!) that were taken care of in the revised version.

Minor Comments

1. It will be suggested that the authors explicitly clarify that ProtGPT2 was not trained on any protein function, and thus the validation presented is limited to biochemical or structural local information or folds and high-level structural characteristics (e.g., disorder regions).
2. The examples in the figures (manuscript Figs 3, 4) are, of course, excellent illustrations of a few cases and thus lack generality. Thus, it is better to include the results in Fig. S3 and S4B as part of the analyses in the main body of the manuscript.
3. I believe that some sentences are more suitable in a 'review or perspective' than in a research paper (e.g. ... ProtGPT2 has learned to speak the protein language../ .. help us understand and 'speak fluently' the protein language...). Also, too general sentences like "protein design has an enormous potential to tackle many current challenges". I fully agree with the statement, but being more specific in describing the tasks addressed in this research is beneficial.
4. While dynamics and conformational changes in proteins are indeed important characteristics, comparing the three repetitions was illustrated. But, please clarify the comparative aspect of protein dynamics.

Reviewer #2:

Remarks to the Author:

The authors did a good job in revising this manuscript. However, there are several problems should be fixed before it can be considered for publication. 1. The unsupervised language models are used in the current study. However, as discussed in the introduction section, there are several other language models, which have been applied to the field of bioinformatics, such as the supervised language models introduced in BioSeq-BLM. It would be interesting and important to discuss the differences between the unsupervised and supervised language models. 2. It is correct to predict the disordered proteins so as to check the quality of the generated proteins by using ProtGPT2's. However, I suggest the authors to analyze the disordered proteins at residue level as well. In other word, I suggest the authors to predict the detailed disordered residues in proteins by using the state-of-the-art methods at residue level, such as DeepIDP-2L. Furthermore, the IUPred3 employed by this study would fail to predict the disordered residues for short proteins,

leading to inaccurate results.

We are thankful to the reviewers for their suggestions and comments. We show our answers in blue.

REVIEWER COMMENTS

Reviewer #1 (Remarks to the Author):

Review (Rebuttal)

In 'design' papers, a legitimate norm is to use some of the prediction for experimental validation (as in many excellent studies, including Anishchenko et al. (2021, on DL hallucination) and others. With a lack of experimental validation (which is hard 'but convincing), generalization of the presented tools is done by setting some statistical evidence for "success". Such measures seek the shortest distance between unseen, new, and known sequences and structures. Such validation is, of course, important and a critical step towards improving models in the protein design field.

The main shortcomings of the original manuscript were the lack of validation and the somewhat deficient novelty. In this version, the authors expanded their tests by testing "novel" sequences statistics versus predicted features (e.g., 2D structure, as presented by AlphaFold2). In the current version, the authors addressed the major comments by providing new set of validations (not for protein functions but for a mode of structurally oriented tasks). The addition of Rosetta Relax scores (RRS) and the extension of AlphaFold prediction to natural and random sequences (20k) are critical additions to improve the manuscript. The other addition concerns testing the protein dynamics (using MD simulations). Also, it suffered from some missing references to background knowledge (that is changing very fast!) that were taken care of in the revised version.

Minor Comments

1. It will be suggested that the authors explicitly clarify that ProtGPT2 was not trained on any protein function, and thus the validation presented is limited to biochemical or structural local information or folds and high-level structural characteristics (e.g., disorder regions).

This is a good idea; we have now added an explanation in the manuscript. We made clear that the sequences in the training and validation sets are non-annotated and that we focus on structural/biochemical validations.

2. The examples in the figures (manuscript Figs 3, 4) are, of course, excellent illustrations of a few cases and thus lack generality. Thus, it is better to include the results in Fig. S3 and S4B as part of the analyses in the main body of the manuscript.

We have moved these results to the main manuscript, they are now shown in Figure 3.

3. I believe that some sentences are more suitable in a 'review or perspective' than in a research paper (e.g. ... ProtGPT2 has learned to speak the protein language.. / .. help us understand and 'speak fluently' the protein language...). Also, too general sentences like "protein design has an enormous potential to tackle many current challenges". I fully agree with the statement, but being more specific in describing the tasks addressed in this research is beneficial.

We agree with these points. We have removed the expression 'speak' the protein language from several locations in the text. Besides, we now mention a few examples to show the potential of protein design in environmental and biomedical fields.

4. While dynamics and conformational changes in proteins are indeed important characteristics, comparing the three repetitions was illustrated. But, please clarify the comparative aspect of protein dynamics.

Thanks for this remark. We have now clarified the reasoning behind these comparisons.

Reviewer #2 (Remarks to the Author):

The authors did a good job in revising this manuscript. However, there are several problems should be fixed before it can be considered for publication.

1. The unsupervised language models are used in the current study. However, as discussed in the introduction section, there are several other language models, which have been applied to the field of bioinformatics, such as the supervised language models introduced in BioSeq-BLM. It would be interesting and important to discuss the differences between the unsupervised and supervised language models.

This is a good point. We had briefly defined in the first results section unsupervised learning, but now we move these two definitions to the Introduction section, including mentioning the remarkable large collection of supervised models in BioSeq-BLM.

2. It is correct to predict the disordered proteins so as to check the quality of the generated proteins by using ProtGPT2's. However, I suggest the authors to analyze the disordered proteins at residue level as well. In other word, I suggest the authors to predict the detailed disordered residues in proteins by using the state-of-the-art methods at residue level, such as DeepIDP-2L. Furthermore, the IUPred3 employed by this study would fail to predict the disordered residues for short proteins, leading to inaccurate results.

We agree with reviewer 2 that it has been shown that short intrinsically disordered regions (SDR, less than 30 amino acids in length) show different properties compared to longer disordered regions (LDR). Tools like DeepIDP-2L that take these differences into account would result in slightly more accurate predictions. However, since we intended to report an overall propensity for disorder across datasets, we believe that the general tendency would be similar using other prediction tools. Besides, the amount of disorder agrees well with the data reported for bacteria and archaea, as pointed out in the manuscript.

Our disorder and globular analysis also included running IUPred3 at the amino acid level, and these results are shown in Table 2. While we know that DeepIPD-2L was explicitly trained for SDR and LDR, we believe that our analysis of choice fits our primary goal, which is to compare the globularity of the sequences in the natural and ProtGTP2-generated datasets. We are thankful for these suggestions and clarify these shortcomings in the text.